# Disease misclassification in electronic healthcare database studies: Deriving validity indices—A contribution from the ADVANCE project

**Kaatje Bollaerts**[1]*, **Alexandros Rekkas**[1,2], **Tom De Smedt**[1], **Caitlin Dodd**[2], **Nick Andrews**[3], **Rosa Gini**[4]

**1** P95 Epidemiology and Pharmacovigilance, Leuven, Belgium, **2** Erasmus Medical Centre Rotterdam, Rotterdam, Netherlands, **3** Statistics, Modelling, and Economics Department, Public Health England, Colindale, London, United Kingdom, **4** Agenzia regionale di sanità della Toscana, Florence, Italy

* Kaatje.Bollaerts@p-95.com

**Data Availability Statement:** All data is based on simulations and can be recalculted using the supplied web application.

## Abstract

There is a strong and continuously growing interest in using large electronic healthcare databases to study health outcomes and the effects of pharmaceutical products. However, concerns regarding disease misclassification (i.e. classification errors of the disease status) and its impact on the study results are legitimate. Validation is therefore increasingly recognized as an essential component of database research. In this work, we elucidate the inter-relations between the true prevalence of a disease in a database population (i.e. prevalence assuming no disease misclassification), the observed prevalence subject to disease misclassification, and the most common validity indices: sensitivity, specificity, positive and negative predictive value. Based on this, we obtained analytical expressions to derive all the validity indices and true prevalence from the observed prevalence and any combination of two other parameters. The analytical expressions can be used for various purposes. Most notably, they can be used to obtain an estimate of the observed prevalence adjusted for outcome misclassification from any combination of two validity indices and to derive validity indices from each other which would otherwise be difficult to obtain. To allow researchers to easily use the analytical expressions, we additionally developed a user-friendly and freely available web-application.

## 1. Introduction

Epidemiology relies on accurately capturing the disease status of subjects within a certain population. Inaccuracies in obtaining the disease status might (strongly) bias the epidemiological findings. Particularly electronic healthcare record (eHR) databases, which have become a prominent source of information in pharmacoepidemiology, are prone the disease misclassification. eHR databases capture healthcare provided to large populations, their size permits the study of rare events and their establishment within clinical practices enables studying real-

**Funding:** Finanacial Disclosure: This research was funded by the Innovative Medicines Initiative (IMI) Joint Undertaking through the ADVANCE project [ 115557]. The IMI is a joint initiative (publicprivate partnership) of the European Commission and the European Federation of Pharmaceutical Industries and Associations (EFPIA) to improve the competitive situation of the European Union in the field of pharmaceutical research. The IMI provided support in the form of salaries for KB, TDS, CD and RG but did not have any additional role in the study design, data collection and analysis, decision to publish, or preparation of the manuscript. AR and NA did not receive any financial compensation for their contribution to this research. The specific roles of the authors are articulated in the 'author contributions' section.

**Competing interests:** Competing Interests statement: This work was funded by the Innovative Medicines Initiative (IMI) Joint Undertaking through the ADVANCE project [ 115557]. P95 Epidemiology and Pharmacovigilance was one of the beneficiaries among the many public partners of this IMI project, including both commercial and non-commercial organisations. P95 did not fund this study and the web-application is made freely available. The IMI provided support in the form of salaries for KB, TDS, CD and RG. This does not alter our adherence to PLOS ONE policies on sharing data and materials. There are no patents, products in development or marketed products associated with this research to declare.

world effects of pharmaceutical products in a timely and cost-efficient manner. However, although eHR databases provide a valuable source of data for pharmacoeopidemiological research, these data are collected primarily for clinical and administrative use rather than for research and as such, concerns regarding data quality exist [1, 2].

Research using eHR databases relies on case-finding algorithms (CFAs), by which subjects captured by the database are classified as diseased or non-diseased, without additional contact with them. The accuracy of the CFA to classify patients depends on the database quality and completeness, the disease of interest and the patient group being studied [3]. Validation of the CFAs, by which the CFA classifications are compared to a reference standard (e.g. chart review, register), is increasingly considered an essential component of eHR database research [3–5]. The validity of the CFAs can be measured by different validity indices; the most commonly used ones are sensitivity (*SE*), specificity (*SP*) positive and negative predictive value (*PPV* and *NPV*). Once the values of such validity indices are known, the observed prevalence or risk estimates can be corrected for misclassification [6, 7].

Despite being considered essential, validation studies are rarely performed because they are very time- and resource intensive [3]. On top, most validation studies only report on *SE* and *PPV* as validation cohorts often do not include subjects without the disease (bench). In this paper, we show how validity indices can be analytically derived from each other.

## 2. Methods

### 2.1. Definitions

A CFA is typically validated by comparing its classifications with that of a reference standard. When the reference standard is assumed to perfectly represent the true dichotomous disease status (i.e. the reference standard is error-free), it is also called the 'gold standard'. The validation data is conventionally captured in a 2 x 2-table representing the joint probability distribution of the CFA-derived classification and the 'gold standard' (Table 1). In this representation, *SE* is the proportion of patients with the disease of interest who are CFA-positive, *SP* is the proportion of persons without the disease who are CFA-negative, *PPV* is the proportion of CFA-positive patients who have the disease of interest and *NPV* is the proportion of CFA-negative persons without the disease of interest. These four validity indices are all conditional probabilities, where *SE*, *SP*, *PPV* and *NPV* are conditioned on the number of diseased, non-diseased, CFA-positives and CFA-negatives, respectively (Table 1). The observed prevalence (*P*) is then the proportion of CFA-positives and the true prevalence ($\pi$) the proportion of diseased among all N subjects. Obtaining the true prevalence is not always possible and requires an error-free test. Note that the observed prevalence and the four validity indices are all CFA-dependent.

**Table 1. Validity indices for dichotomous data: Sensitivity (SE), specificity (SP) positive (PPV) and negative predictive value (NPV) the observed (P) and true prevalence ($\pi$).**

| | | 'Gold' standard | | |
|---|---|---|---|---|
| | | Positive | Negative | Validity index |
| Case Finding Algorithm | Positive | Nr. of True positives TP | Nr. of False positives FP | $PPV = \text{TP}/(\text{TP}+\text{FP})$ |
| | Negative | Nr. of False negatives FN | Nr. of True negatives TN | $NPV = \text{TN}/(\text{FN}+\text{TN})$ |
| | Validity index | $SE = \text{TP}/(\text{TP} + \text{FN})$ | $SP = \text{TN}/(\text{FP} + \text{TN})$ | $N = \text{TP}+\text{FP}+\text{FN}+\text{TN}$ $P = (\text{TP}+\text{FP})/N$ $\pi = (\text{TP}+\text{FN})/N$ |

## 2.2. Interrelationships between validity indices

The 2 x 2-table representation (Table 1) shows how the true prevalence, observed prevalence and the validity indices SE, SP, PPV and NPV are interrelated. Alternatively, these interrelations can be expressed in terms of the actual parameters themselves (and not the cell counts of the 2x2-table). Indeed, starting from the expression relating the observed prevalence to the true prevalence[7, 8] and from the definitions of PPV and NPV [9], we have the following system of algebraic equations with six unknown parameters;

$$P = SE\,\pi + (1 - SP)(1 - \pi), \tag{1}$$

$$PPV = SE\,\pi/(SE\,\pi + (1 - SP)(1 - \pi)), \tag{2}$$

$$NPV = SP(1 - \pi)/((1 - SE)\pi + SP(1 - \pi)). \tag{3}$$

Hence, if we know three parameters, we can derive the others. The observed prevalence $P$ is easily obtained by applying the CFA to the population in the database. Then, once we input two other parameters, the remaining parameters can be analytically derived by solving the system of algebraic equations above. For all combinations of $P$ and any two other parameters, the analytical solutions for the remaining three parameters are given in Table 2.

The true prevalence, observed prevalence and the four validity indices are all (conditional) probabilities, and hence are bounded between zero and one. This imposes constraints on the input parameters without which the analytically derived parameters might be outside the zero-to-one range (constraints in S1 Table). More restrictive constraints result if we impose that the CFA should detect disease better than chance alone [7] (constraints in S2 Table). A CFA performs better than chance if it selects diseased persons with a higher probability than it does non-diseased persons. Note that the issue of a CFA performing worse than chance is easily alleviated through swapping the CFA-results, i.e. by re-labeling the CFA-positive results as negative and vice versa.

Finally, if the uncertainty associated with some of the input parameters is known, the uncertainty can be propagated to the derived parameters through Monte Carlo (MC) sampling. In this process, repeated samples from the statistical distributions of the input parameters are drawn. As the input parameters are all probabilities, it is naturally to assign beta distributions to them[10]. Then, for each MC sample of three input parameters, the remaining

**Table 2. Overview of the interrelations between validity indices and the true prevalence, given the observed prevalence $P$ and two other parameters.**

| | Known | Expressions | | |
|---|---|---|---|---|
| 1. | $\Pi$, P, SE | $SP = 1 - \frac{(P - SE \times \Pi)}{1 - \Pi}$ | $PPV = \frac{SE \times \Pi}{P}$ | $NPV = 1 - \frac{\Pi(1 - SE)}{1 - P}$ |
| 2. | $\Pi$, P, SP | $SE = \frac{P - (1 - \Pi)(1 - SP)}{\Pi}$ | $PPV = 1 - \frac{(1 - \Pi)(1 - SP)}{P}$ | $NPV = \frac{SP\ (1 - \Pi)}{1 - P}$ |
| 3. | $\Pi$, P, PPV | $SE = \frac{P \times PPV}{\Pi}$ | $SP = 1 - \frac{P(1 - PPV)}{1 - \Pi}$ | $NPV = 1 - \frac{\Pi - P \times PPV}{1 - P}$ |
| 4. | $\Pi$, P, NPV | $SE = 1 - \frac{1 - \Pi - NPV(1 - P)}{\Pi}$ | $SP = \frac{NPV(1 - P)}{1 - \Pi}$ | $PPV = 1 - \frac{1 - \Pi - NPV(1 - P)}{P}$ |
| 5. | P, SE, SP | $\Pi = \frac{P + SP - 1}{SE + SP - 1}$ | $PPV = 1 - \frac{(P - SE)(1 - SP)}{P\ (1 - SP - SE)}$ | $NPV = \frac{(P - SE)\ SP}{(1 - P)(1 - SP - SE)}$ |
| 6. | P, SE, PPV | $\Pi = \frac{P \times PPV}{SE}$ | $SP = 1 - \frac{P\ (1 - PPV)SE}{SE - P \times PPV}$ | $NPV = 1 - \frac{(1 - SE)\ (P \times PPV)}{SE\ (1 - P)}$ |
| 7. | P, SE, NPV | $\Pi = \frac{(1 - P)(1 - NPV)}{1 - SE}$ | $SP = \frac{(1 - P)(1 - SE)\ NPV}{(1 - SE) - (1 - P)(1 - NPV)}$ | $PPV = \frac{SE \times (1 - P)(1 - NPV)}{P(1 - SE)}$ |
| 8. | P, SP, PPV | $\Pi = 1 - \frac{P \times (1 - PPV)}{1 - SP}$ | $SE = \frac{P \times PPV(1 - SP)}{1 - SP - P(1 - PPV)}$ | $NPV = \frac{P \times SP \times (1 - PPV)}{(1 - P)(1 - SP)}$ |
| 9. | P, SP, NPV | $\Pi = 1 - \frac{(1 - P) \times NPV}{SP}$ | $SE = \frac{P \times SP - (1 - SP)(1 - P)\ NPV}{SP - (1 - P) \times NPV}$ | $PPV = \frac{P \times SP - (1 - SP)(1 - P)NPV}{P \times SP}$ |
| 10. | P, PPV, NPV | $\Pi = (1 - P)(1 - NPV) + P \times PPV$ | $SE = \frac{P \times PPV}{(1 - P)(1 - NPV) + P \times PPV}$ | $SP = \frac{(1 - P) \times NPV}{1 - (P \times PPV + (1 - P)(1 - NPV))}$ |

parameters are derived. This results in a distribution of derived parameters, based on which uncertainty intervals (UIs) can be derived [11]. As the true prevalence, observed prevalence and the validity indices are correlated, the MC sampling should ideally reflect this. Not accounting for correlation among the parameters might result in too wide UIs and in sampling parameter combinations that violate the constraints above. However, the correlations among the parameters are typically unknown. Therefore, we used independent sampling but rejected the invalid parameter combinations as defined by the constraints in S1 Table or S2 Table.

## Web-application

To allow users to easily explore the interrelations between the true prevalence, observed prevalence and the validity indices SE, SP, PPV and NPV, we developed a web application using R [12] and the Shiny package [13]. The application is available from https://apps.p-95.com/Interr/. The application calculates the validity indices given user-defined values of the observed prevalence and any other two parameters. Optionally, the 95% percentile UIs of the derived parameters are calculated through MC simulation when the 95% confidence intervals (CIs) of the known parameters are provided. More specifically, we assign beta distributions to all known parameters for which CIs are provided, with the shape parameters of the beta distribution derived from the provided mean values and CIs based on the method of moments [14]. Invalid combinations of parameter values are discarded and the percentages of constraint violations are reported. We provide two types of UIs, one with the 'bounded between 0 and 1' constraints applied (S1 Table) and one with the more restrictive 'better than chance' constraints applied (S2 Table)

To demonstrate the web-application, we used published results on the validation of two CFAs, one for intussusception and one for pneumonia, and derived any three indices using the other two as input parameters.

## 2.4. Sensitivity analyses

We additionally conducted sensitivity analyses to investigate the impact of estimation error in the input parameters on the derived parameters. For every combination of the observed prevalence and any two other parameters, we varied the input parameters one-at-the-time (OAT) while keeping the remaining input parameters at their baseline values [15]. Specifically, the input parameters $p$ are varied between an under- and an overestimation with one standard error s.e. (i.e. between $p - s.e.$ and $p + s.e.$) with s.e. calculated for the binomial proportion $p$ from a sample of size 1000. We investigated three baseline scenarios for varying levels of $\pi = \{0.01, 0.05, 0.2\}$ while keeping SE and SP fixed at 0.75 and 0.99, respectively. The corresponding baseline values for the observed prevalence and the predictive values were $P = \{0.02, 0.05, 0.16\}$, $PPV = \{0.43, 0.80, 0.95\}$ and $NPV = \{1.0, 0.99, 0.94\}$. The biases of the derived indices are expressed relative to their standard errors as well. For the sensitivity analyses, we applied the less restrictive 'bounded between 0 and 1' constraints.

## 3. Results

### 3.1. Illustrations

Ducharme et al conducted a validation study of the diagnostic, procedural, and billing codes for the identification of intussusception in children <18 years living in the Census Metropolitan Area of Ottawa (Ontario, Canada) between 1995 and 2010 [16]. The authors calculated SE, SP, PPV, and NPV using manual validation of hospital records using the Brighton Collaboration diagnostic criteria as a gold standard. Case finding algorithms were based on a single or

combination of ICD-9 diagnosis codes, procedure codes, and billing codes. Among the 417,997 patients, 185 patients (0.044%) met the case criteria according to the CFA chosen by the authors and 150 patients (0.036%) where intussusception cases. The CFA's PPV was 72.4% (95%CI: 65.4–78.7) and the SE was 89.3% (95% CI: 83.3–93.8), while both the NPV and the SP were >99.9% (95% CI: >99.9–100.0). Starting from the observed prevalence, SE and PPV, we derived the NPV and SP (Fig 1). The derived values for SP and NPV were the same as those reported in the paper. The true prevalence was derived to be 0.036% (95% UI: 0.034–0.038), equal to the study estimate. Starting from the observed prevalence, the PPV and the true prevalence led to a SE of 88.5% (84.4–92.6), close to the study estimate of 89.3%.

A second example was the validation study of claims-based pneumonia CFA. In a cross-sectional study of patients visiting the emergency department (ED) of a hospital in Salt Lake City, Utah during a 5-month period, Aronsky et al assessed the validity of five different claims-based pneumonia CFA against a 'gold standard' of manual review of each patient encounter [17]. Among 10828 ED encounters, 272 (2.51%) were cases of pneumonia according to the 'gold standard'. Their selected algorithm was positive for 219 encounters (2.02%). For this algorithm, the authors reported SE of 65.1% (95% CI: 59.2–70.5), SP of 99.6% (95% CI: 99.5–99.7), PPV of 80.8% (95% CI: 75.1–85.5), and NPV of 99.1% (95% CI: 98.9–99.3). First, we used as input the PPV and NPV. The derived SE and SP were the same as those reported in the paper, as well as the true prevalence (2.51%; 95% UI:2.4–2.6) (Fig 2). Second, we used PPV and an interval for the true prevalence (2.00–3.00%) as input parameters. The derived ranges for SE, SP and NPV were [54.4–81.6], [99.6–99.6] and [98.6–99.6]; all including the originally reported values.

### 3.2. Sensitivity analyses

The impact of changing the input parameters (from -1 s.e. to + 1 s.e.) on the output parameters is depicted by the vertical bars in Figs 3 and 4. The biases of the derived indices are expressed relative to their standard errors as well and are truncated at ±3 s.e. For example, for the input parameter combination $\pi - P - SE$ and when $\pi = 0.01$ (Fig 3: upper left panel), varying $\pi$ from

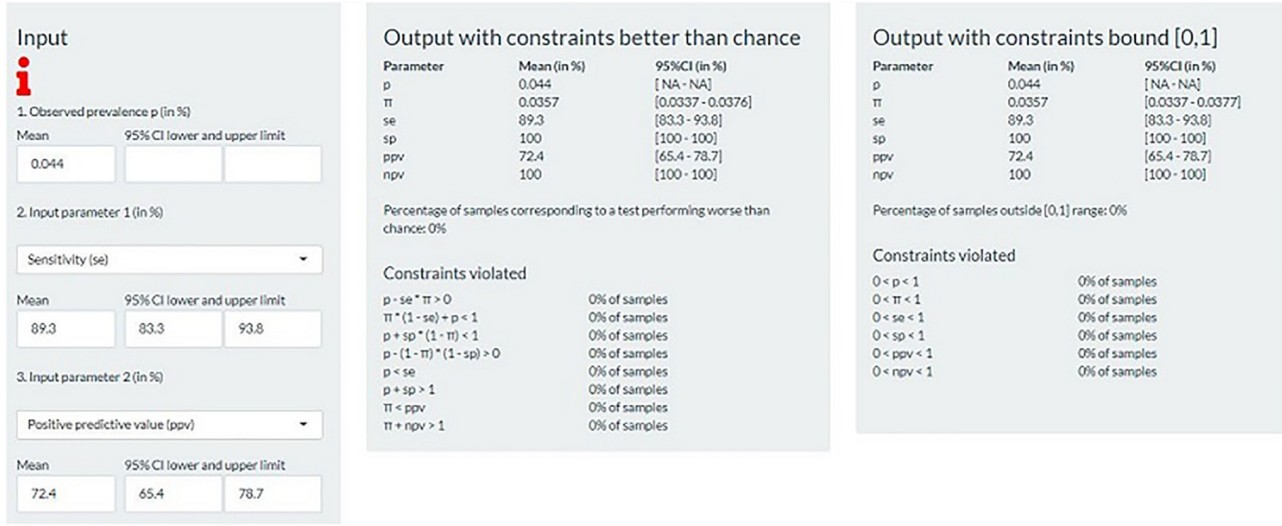

**Fig 1. Intussusception; deriving true prevalence, specificity and negative predictive value from the observed prevalence, sensitivity and positive predictive value.**

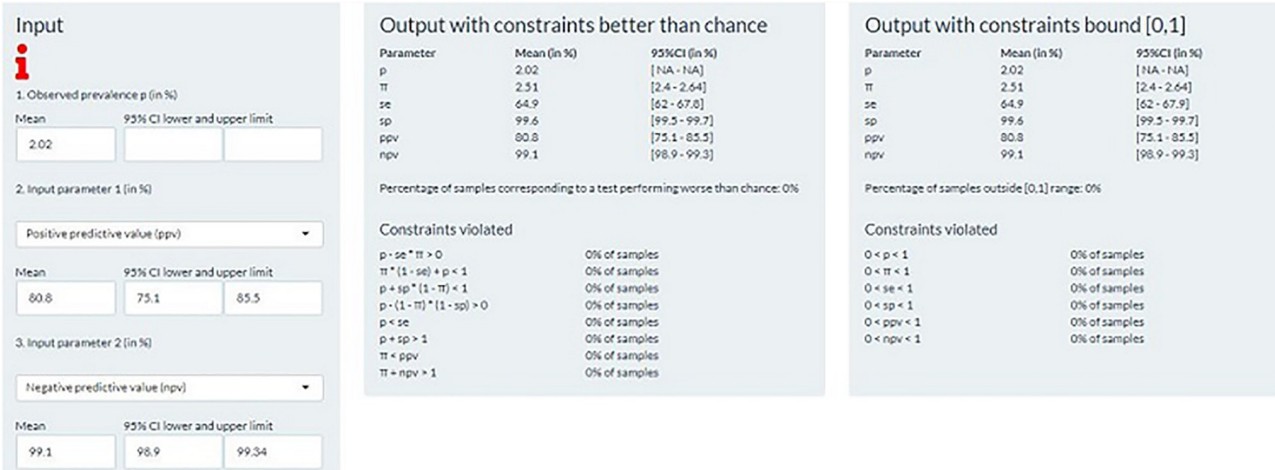

**Fig 2. Pneumonia; deriving true prevalence, sensitivity and specificity from the observed prevalence, positive and negative predictive value.**

-1 s.e. to + 1 s.e, has a small impact on SE and NPV ($<$ 1 s.e. change in both directions), but a more substantial impact on PPV (~2 s.e. change in both directions). The combined results indicate that for the scenarios investigated the estimation error of the derived parameters is smallest when using the parameter combination *P*–SE–PPV.

## 4. Discussion

Starting from the interrelations between the true disease prevalence, the observed prevalence (as estimated from the misclassified data) and the four validity indices *SE*, *SP*, *PPV* and *NPV*, we derived the analytical expressions (formulas) to obtain for every combination of the observed prevalence and two other parameters the remaining three parameters. To facilitate the use of these analytical expressions, we developed a freely available user-friendly web-application.

The analytical expressions and web-application can be used for various purposes. First, they can be used to adjust a prevalence estimate for outcome misclassification. The expression to derive the true prevalence from the observed prevalence, SE and SP was already published in the late 70's, and known as the Rogan-Gladen estimator [7]. Our application allows users to obtain an estimate of the true prevalence given an estimate of the observed prevalence and any two other validity indices. These expressions were previously used to adjust *Bordetella Pertussis* incidence rates from five European healthcare databases for outcome misclassification [18].

To the best of our knowledge, none of these analytical expressions were prior available besides the Rogan-Gladen estimator. Second, the analytical expressions can be used to derive validity indices that are otherwise difficult to obtain. Particularly SP and NPV require very large validation studies, especially in the case of rare diseases. Benchimol et al [3] conducted a systematic review of validation studies of CFAs and found that only 36.9% of the studies reported four or more validity indices. They found that the most common validity indices used to report the diagnostic accuracy of CFAs are SE (67.2%) and PPV (63.8%) and to a lesser extent SP (49.8%) and NPV (32.1%). Another review study found that most studies that validate diagnoses in the Clinical Practice Research Database (CPRD) were restricted to assessing the proportion of CFA-positive cases that were confirmed by medical record review or responses to questionnaires [19, 20], thus only providing an estimate of PPV whereas at least two validity indices are required to adjust a prevalence estimate for outcome misclassification.

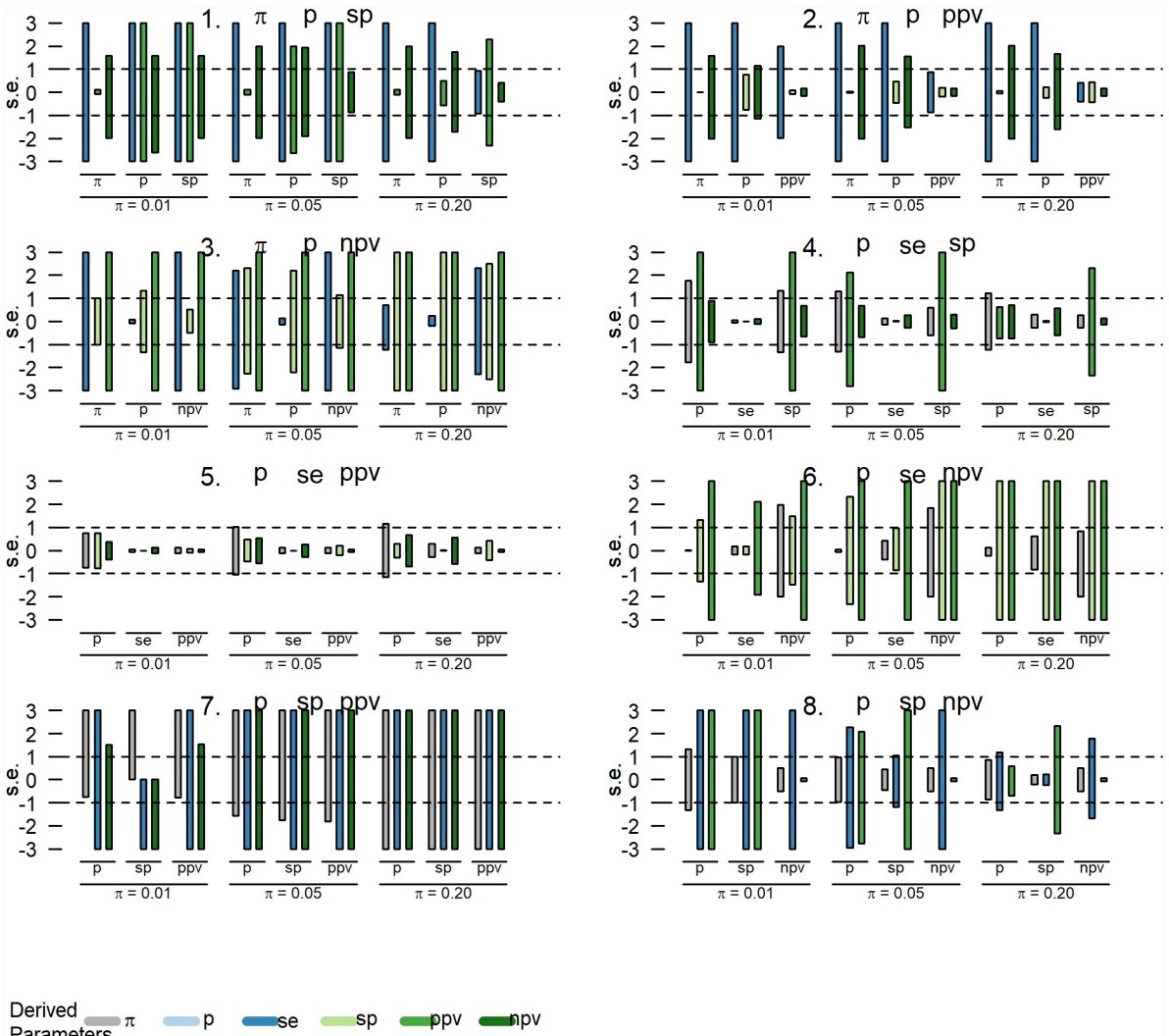

**Fig 3. Results of the sensitivity analyses: Investigating the impact of changing the input parameters from -1 to +1 standard error (s.e.) on the derived parameters for varying levels of true prevalence, $\pi$ = {0.01, 0.05, 0.2}, SE = 0.95 and SP = 0.75.** The bias of the derived indices are truncated at ±3 s.e.

In such cases where only one validity index is reported, the remaining validity indices can be derived when an estimate of the true prevalence is available. Such an estimate of the true prevalence might be obtained from external data sources such as disease registers or national surveillance systems. Obviously, in this case, it is important to ensure that the external estimate applies to the database population under study. Third, the comparison of validation studies is often hampered by the use of different validity indices. The ability to convert indices will facilitate this comparison. Fourth, the possibility to independently estimate different validity indices using different validation samples (e.g. a sample of diseased subjects to estimate SE and another sample of CFA-positives to estimate PPV) will make validation more feasible. It will undoubtedly reduce the sample size requirements compared to a comprehensive validation study by which the 'gold standard' measure is obtained for a random sample of the database population. Especially for rare diseases, such validations studies are unfeasible as very large sample sizes are required to capture at least some diseased subjects.

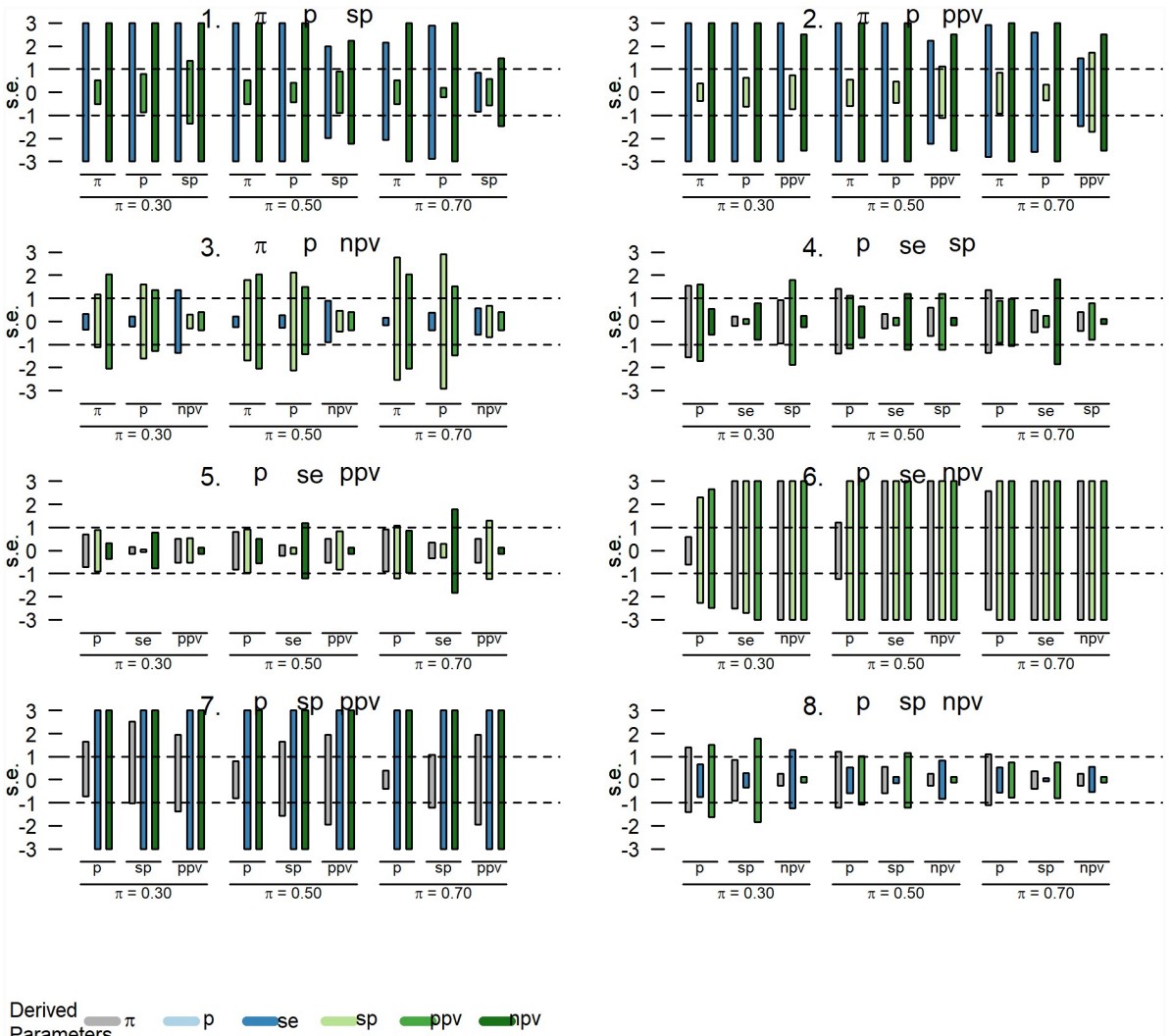

**Fig 4. Results of the sensitivity analyses: Investigating the impact of changing the input parameters from -1 to +1 standard error (s.e.) on the derived parameters for varying levels of true prevalence, π = {0.3, 0.5, 0.7}, SE = 0.95 and SP = 0.75.** The bias of the derived indices are truncated at ± 3 s.e.

The methodology of analytically deriving validity indices has limitations. The presence of sampling error or selection bias might result in invalid parameter combinations (i.e. resulting in derived parameters outside the [0,1] range or corresponding to a CFA that performs worse than chance). To investigate the impact of estimation error in the input parameters on the derived parameters we conducted sensitivity analyses. The results show that, for the scenarios we investigated, the parameter combinations *P*–SE–PPV resulted in the smallest estimation errors in the derived parameters. The assumptions applying to our analytical derivations are the same as those underlying the conventional 2 x 2-table representation of validity indices (Table 1). These assumptions are that the true disease status is truly dichotomous and the dichotomous 'gold standard' measure reflects the true disease status without error. However, disease is not always absent or present and there might be an underlying continuous condition (i.e. spectrum of severity) on which classification of disease status is based, varying from the clear absence to the clear presence of disease. In such cases, the SE and SP depend on the

distribution of the underlying condition, and hence on the true disease prevalence [20, 21]. On top, if the gold standard measure is erroneous, the validity indices will be biased [21]. The methodology applies to prevalence estimates and incidence proportions, and not to the more commonly used incidence rate. Also, and irrespective of the validation methodology used, the validity of CFAs might depend on many factors such as population characteristics, access to healthcare and the completeness of the medical information contained in the database, thereby limiting the generalizability of the validity indices to populations others than those for which the validity of the CFA was initially assessed [2, 19]. Finally, disease misclassification might be differential, meaning that the misclassification depends on the exposure status, which leads to biased estimates of the exposure-disease association in both directions [22]. In this case, it is important to obtain validity indices by exposure status.

Despite these limitations, we echo many others [2, 3, 5] that validation of CFAs is essential to permit proper interpretation of the results obtained from healthcare database studies. The estimated validity indices might ultimately be used to adjust estimates of disease occurrence [7] or risk [6] for misclassification or to adjust power calculations [23]. By providing the analytical expressions regarding the inter-relations of the observed prevalence, true prevalence and the most commonly used validity indices, we hope to contribute to a more widespread use of validation studies and their results.

## Supporting information

**S1 Table. Constraints on the input parameters ensuring that the derived parameters belong to the interval [0,1].**
(DOCX)

**S2 Table. Parameter constraints corresponding to a case-finding algorithm that performs better than chance.**
(DOCX)

## Author Contributions

**Conceptualization:** Kaatje Bollaerts, Alexandros Rekkas, Tom De Smedt, Caitlin Dodd, Nick Andrews, Rosa Gini.

**Formal analysis:** Kaatje Bollaerts.

**Investigation:** Alexandros Rekkas, Tom De Smedt.

**Methodology:** Kaatje Bollaerts.

**Project administration:** Kaatje Bollaerts.

**Software:** Kaatje Bollaerts, Alexandros Rekkas, Tom De Smedt, Rosa Gini.

**Validation:** Kaatje Bollaerts, Rosa Gini.

**Visualization:** Kaatje Bollaerts, Tom De Smedt.

**Writing – original draft:** Kaatje Bollaerts, Tom De Smedt.

**Writing – review & editing:** Kaatje Bollaerts, Alexandros Rekkas, Tom De Smedt, Caitlin Dodd, Nick Andrews, Rosa Gini.

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
