## [Decision Letter · Decision Letter 0]

23 Jan 2020

PONE-D-19-33276

Validation of algorithms in healthcare databases: deriving validity indices from each other

PLOS ONE

Dear Dr Bollaerts,

Thank you for submitting your manuscript to PLOS ONE. After careful consideration, we feel that it has merit but does not fully meet PLOS ONE’s publication criteria as it currently stands. Therefore, we invite you to submit a revised version of the manuscript that addresses the points raised during the review process.

We would appreciate receiving your revised manuscript by Mar 08 2020 11:59PM. To enhance the reproducibility of your results, we recommend that if applicable you deposit your laboratory protocols in protocols.io, where a protocol can be assigned its own identifier (DOI) such that it can be cited independently in the future. For instructions see: http://journals.plos.org/plosone/s/submission-guidelines#loc-laboratory-protocols

We look forward to receiving your revised manuscript.

Kind regards,

Junwen Wang, Ph.D.

Academic Editor

PLOS ONE

Journal Requirements:

2. Please include your tables as part of your main manuscript and remove the individual files. Please note that supplementary tables (should remain/ be uploaded) as separate "supporting information" files

3. We note that the Figures in your submission contain copyrighted images. All PLOS content is published under the Creative Commons Attribution License (CC BY 4.0), which means that the manuscript, images, and Supporting Information files will be freely available online, and any third party is permitted to access, download, copy, distribute, and use these materials in any way, even commercially, with proper attribution. For more information, see our copyright guidelines: http://journals.plos.org/plosone/s/licenses-and-copyright.

a.    You may seek permission from the original copyright holder of your Figures to publish the content specifically under the CC BY 4.0 license.

4. Thank you for stating the following in the Competing Interests section: "The authors have declared that no competing interests exist."

We note that one or more of the authors are employed by a commercial company: P95 Epidemiology and Pharmacovigilance.

6. One of the noted authors is a group or consortium: ADVANCE consortium. In addition to naming the author group, please list the individual authors and affiliations within this group in the acknowledgments section of your manuscript. Please also indicate clearly a lead author for this group along with a contact email address.

Reviewers' comments:

Reviewer's Responses to Questions

**Comments to the Author**

1. Is the manuscript technically sound, and do the data support the conclusions?

Reviewer #1: Partly

Reviewer #2: Yes

Reviewer #3: Yes

2. Has the statistical analysis been performed appropriately and rigorously? 

Reviewer #1: No

Reviewer #2: Yes

Reviewer #3: Yes

3. Have the authors made all data underlying the findings in their manuscript fully available?

Reviewer #1: Yes

Reviewer #2: Yes

Reviewer #3: Yes

4. Is the manuscript presented in an intelligible fashion and written in standard English?

Reviewer #1: Yes

Reviewer #2: Yes

Reviewer #3: Yes

5. Review Comments to the Author

Reviewer #1: This work tried to address the relationship been the performance parameters, e.g. sensitivity, specificity, PPV, NPV, hidden and EMR disease prevalence. I don’t think this work adds any value to the field as it obviously confused the disease prevalence which can be either detected by the EMR or the population. EMR encapsulates data from limited groups of patients, e.g. inpatient, outpatient, and ER settings, therefore, could be under or over report the disease prevalence. I don’t think mathematically you can derive the hidden or true disease prevalence or the case finding algorithm performance by a simple analysis. The only validation approach is to establish a second site prospective trial with robust inclusion/exclusion design. Any case finding algorithm which survives the prospective learning transfer would be regarded as validated. In that regard, I suggest this work to be rejected.

Reviewer #2: Bollaerts et al. validated the algorithms in healthcare databases, and probably more importantly, developed a freely available use-friendly web-application to facilitate the use of their analytical expressions. The manuscript provides important information and method for the community considering the growing need for using real world data to study health outcomes and to study safety and efficacy of pharmaceutical products.

The manuscript is overall well written and comprehensive, I only have minor comments;

1. Provide a very brief how-to-use explanations for the web-application as a supplement.

2. In the Definitions in the method section, add short descriptions for “true prevalence” of how the determinations of the proportion of the diseased usually are performed or obtained. Estimation of the true prevalence is not always available.

3. Some of the labels in the Figure 3 and Figure 4 are not legible, please make sure that they will be.

Reviewer #3: The paper is of great interest, given the increasing use of electronic healthcare records for epidemiological purposes (not restricted to pharmacoepidemiology). The formulas to derive validity indices and the user-friendly web-application are very useful when it is possible to apply them. As specified by the authors their accuracy depends on the presence of a solid and unbiased sample used for validation studies, as well as the presence of a reliable prevalence estimate, from which to derive other parameters.

The paper has been framed in the field of pharmacoepidemiology, although the derivation of validity indices can be applied to other filed, firstly descriptive epidemiology.

The correction for potential nondifferential misclassified outcomes in epidemiological study using derived validity indices is only mentioned in the paper and it should be better discussed with illustration in the field of pharmocoepidemiology.

6. PLOS authors have the option to publish the peer review history of their article (what does this mean?). If published, this will include your full peer review and any attached files.

Reviewer #1: No

Reviewer #2: No

Reviewer #3: No

---

## [Author Response · Author response to Decision Letter 0]

6 Mar 2020

Editor:

[Reply]: We checked the PLOS ONE’s style requirements and formatted accordingly.

2). Please include your tables as part of your main manuscript and remove the individual files. Please note that supplementary tables (should remain/ be uploaded) as separate "supporting information" files.

[Reply]: Tables are included as part of the main manuscript

3) We note that the Figures in your submission contain copyrighted images. All PLOS content is published under the Creative Commons Attribution License (CC BY 4.0), which means that the manuscript, images, and Supporting Information files will be freely available online, and any third party is permitted to access, download, copy, distribute, and use these materials in any way, even commercially, with proper attribution. For more information, see our copyright guidelines: http://journals.plos.org/plosone/s/licenses-and-copyright.

 [Reply]: The authors of the submitted manuscript are the copyright holders and agree with publishing these figures under the CC BY4.0 license

4). Thank you for stating the following in the Competing Interests section: "The authors have declared that no competing interests exist." We note that one or more of the authors are employed by a commercial company: P95 Epidemiology and Pharmacovigilance. Please provide an amended Funding Statement declaring this commercial affiliation, as well as a statement regarding the Role of Funders in your study. If the funding organization did not play a role in the study design, data collection and analysis, decision to publish, or preparation of the manuscript and only provided financial support in the form of authors' salaries and/or research materials, please review your statements relating to the author contributions, and ensure you have specifically and accurately indicated the role(s) that these authors had in your study. You can update author roles in the Author Contributions section of the online submission form.

 [Reply]: This work was funded by the Innovative Medicines Initiative (IMI) Joint Undertaking through the ADVANCE project [№ 115557]. P95 was one of the beneficiaries among the many public partners of this IMI project, including both commercial and non-commercial organisations.P95 did not fund this study and the web-application is made freely available. 

Amended funding statement: This research was funded by the Innovative Medicines Initiative (IMI) Joint Undertaking through the ADVANCE project [№ 115557]. The IMI is a joint initiative (public-private partnership) of the European Commission and the European Federation of Pharmaceutical Industries and Associations (EFPIA) to improve the competitive situation of the European Union in the field of pharmaceutical research. The IMI provided support in the form of salaries for KB, TDS, CD and RG but did not have any additional role in the study design, data collection and analysis, decision to publish, or preparation of the manuscript. AR and NA did not receive any financial compensation for their contribution to this research. 

Competing interest statement: The authors have declared that no competing interests exist.

[Reply]: Figure captions of the Supporting Information files have been included at the end of the manuscript.

6. One of the noted authors is a group or consortium: ADVANCE consortium. In addition to naming the author group, please list the individual authors and affiliations within this group in the acknowledgments section of your manuscript. Please also indicate clearly a lead author for this group along with a contact email address.

[Reply]: It was not meant to have a group authorship but rather acknowledge that this work was carried out under the auspices of the ADVANCE project. Therefore we suggest to omit ‘on behalf of the ADVANCE consortium’ from the author list and change the title as follows ‘Outcome misclassification in electronic healthcare database studies: deriving validity indices – A contribution from the ADVANCE project’. 

Reviewer 1: 

This work tried to address the relationship been the performance parameters, e.g. sensitivity, specificity, PPV, NPV, hidden and EMR disease prevalence. I don’t think this work adds any value to the field as it obviously confused the disease prevalence which can be either detected by the EMR or the population. EMR encapsulates data from limited groups of patients, e.g. inpatient, outpatient, and ER settings, therefore, could be under or over report the disease prevalence. I don’t think mathematically you can derive the hidden or true disease prevalence or the case finding algorithm performance by a simple analysis. The only validation approach is to establish a second site prospective trial with robust inclusion/exclusion design. Any case finding algorithm which survives the prospective learning transfer would be regarded as validated. In that regard, I suggest this work to be rejected. 

[Reply]: Our paper does not address the issue of a potential lack of representativeness of the electronic healthcare database population for the population of interest. Instead, our paper aims to contribute to the issue of disease misclassification in database research. The title and the abstract has been modified to make this explicit.

Reviewer 2: 

Reviewer #2: Bollaerts et al. validated the algorithms in healthcare databases, and probably more importantly, developed a freely available use-friendly web-application to facilitate the use of their analytical expressions. The manuscript provides important information and method for the community considering the growing need for using real world data to study health outcomes and to study safety and efficacy of pharmaceutical products.

The manuscript is overall well written and comprehensive, I only have minor comments;

1. Provide a very brief how-to-use explanations for the web-application as a supplement.

[Reply]: We developed a short user-manual which can be accessed from the link to the application https://apps.p-95.com/Interr/.

2. In the Definitions in the method section, add short descriptions for “true prevalence” of how the determinations of the proportion of the diseased usually are performed or obtained. Estimation of the true prevalence is not always available.

[Reply]: We added the following sentence in the methods section: “Obtaining the true prevalence is not always possible and requires an error-free test.”

3. Some of the labels in the Figure 3 and Figure 4 are not legible, please make sure that they will be.

[Reply]: The labels of the figures have been enlarged to the extent possible.

Reviewer 3: 

The paper is of great interest, given the increasing use of electronic healthcare records for epidemiological purposes (not restricted to pharmacoepidemiology). The formulas to derive validity indices and the user-friendly web-application are very useful when it is possible to apply them. As specified by the authors their accuracy depends on the presence of a solid and unbiased sample used for validation studies, as well as the presence of a reliable prevalence estimate, from which to derive other parameters.

The paper has been framed in the field of pharmacoepidemiology, although the derivation of validity indices can be applied to other filed, firstly descriptive epidemiology.

[Reply]: The first paragraph of the introduction has been modified to give a broader context to the methodology.

The correction for potential nondifferential misclassified outcomes in epidemiological study using derived validity indices is only mentioned in the paper and it should be better discussed with illustration in the field of pharmocoepidemiology.

[Reply]: The correction non-differential outcomes is described in the second paragraph of the discussion. We also add, as part of the paragraph mentioning the limitations, that disease misclassification can also be differential, and in this case, requires validity indices by exposure status.

---

## [Editor Report · Decision Letter 1]

23 Mar 2020

Disease misclassification in electronic healthcare database studies: deriving validity indices – A contribution from the ADVANCE project

PONE-D-19-33276R1

Dear Dr. Bollaerts,

We are pleased to inform you that your manuscript has been judged scientifically suitable for publication and will be formally accepted for publication once it complies with all outstanding technical requirements.

With kind regards,

Junwen Wang, Ph.D.

Academic Editor

PLOS ONE
---

## [Editor Report · Acceptance letter]

9 Apr 2020

PONE-D-19-33276R1 

Disease misclassification in electronic healthcare database studies: deriving validity indices – A contribution from the ADVANCE project 

Dear Dr. Bollaerts:

I am pleased to inform you that your manuscript has been deemed suitable for publication in PLOS ONE. Congratulations! Your manuscript is now with our production department. 

With kind regards,

on behalf of

Dr. Junwen Wang 

Academic Editor

PLOS ONE